# Direct visual observation of pedal motion-dependent flexibility of single covalent organic frameworks

Hongbin Chi[1], Yang Liu[1,2], Ziyi Li[1], Wanxin Chen[1] & Yi He ®[1] ✉

Flexible covalent organic frameworks (COFs) have been studied for applications containing sorption, selective separation, and catalysis. How to correlate the microscopic structure with flexibility in COFs is a great challenge. Herein, we visually track the flexible deformation behaviors of single COF-300 and COF-300-AR particles in response to solvent vapour guests with dark-field microscopy (DFM) in an in operando manner. COF-300-AR with freely-rotating C-N single bonds are synthesized by the reduction of imine-based COF-300 consisting of rigid C=N double bonds without changing topological structure and crystallinity. Unexpectedly, we observe that the flexible deformation of COF-300 is extremely higher than that of COF-300-AR despite it bears many C-N single bonds, clearly illustrating the apparent flexibility decrease of COF-300 after reduction. The high spatiotemporal resolution of DFM enables the finding of inter-particle variations of the flexibility among COF-300 crystals. Experimental characterizations by variable-temperature X-ray diffraction and infrared spectroscopy as well as theoretical calculations demonstrate that the flexible deformation of COF-300 is ascribed to the pedal motion around rigid C=N double bonds. These observations provide new insights into COF flexibility.

Covalent organic frameworks (COFs) have shown great application potential in separation, gas storage, catalysis, and sensing because of their tunable active sites, large surface area, good stability, high crystallinity, and regular microporous structures[1–3]. In particular, the flexibility of some COFs is considered as a key feature differing from traditional porous materials, including zeolites and activated carbons[4–9]. For instance, flexible COFs can dynamically change the pore diameters/shapes during the host-guest recognition process, displaying attractive elasticity, good affinity, and self-adaptive ability for superior adsorption, highly selective separation, and catalytic activity improvement[10–12]. To obtain the flexible COFs, flexible building blocks such as C−O and C−N single bonds have been incorporated into the backbone[4,13–16]. Indeed, sp³ carbon-involved single bonds rotate freely, and the rotation of double bonds is strictly restricted. However, it is questionable whether the local flexible single bonds must result in

the whole flexibility of COF crystals. Accordingly, fundamentally understanding the microscopic structure-flexibility relationship is extremely important for guiding the design and development of advanced flexible COFs.

Currently, the flexibility of COFs is usually revealed by X-ray diffractions (XRD) and adsorption isotherms[8,17]. Although adaptive structural transformations such as contraction, expansion, and distortion have been documented[7,8], these bulk-ensemble measurements simply average the flexible behavior of many COF crystals due to the lack of spatial resolution, masking the flexibility differences among individual COF particles. Actually, the preparation of high-quality crystals is a still huge challenge in the chemistry of COFs[18]. The synthesized flexible COFs always display a wide distribution in regard to shape, size, crystallinity, and surface defect[7,19]. Significant heterogeneity among different flexible COF crystals is inevitable.

[1]School of Nuclear Science & Technology, Southwest University of Science and Technology, 621010 Mianyang, P. R. China. [2]Sichuan College of Architectural Technology, 618000 Deyang, Sichuan, P. R. China. ✉e-mail: yhe2014@126.com

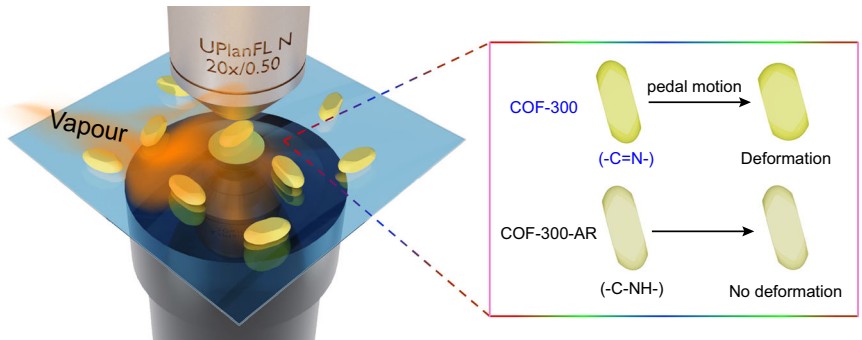

**Fig. 1 | Imaging of pedal motion-dependent flexibility in single COFs.** Schematic illustration of DFM setup for tracking the flexible deformations of single COF-300 and COF-300-AR in the presence of solvent vapours.

Investigating the structural transformation of flexible single COF particles therefore represents a powerful avenue to gain the intrinsic flexibility via eliminating the heterogeneity.

Dark-field microscopy (DFM) has emerged as an economical and effective optical imaging technique for in situ monitoring of physical properties and chemical processes with nanoscale spatial resolution in the past decades[20]. DFM which collects the elastic scattering light has been widely applied to image individual plasmonic nanomaterials and dielectric nanoparticles[21–23]. Recently, we have confirmed the feasibility of DFM for spatiotemporally investigating a series of physical and chemical processes of single COFs and metal-organic frameworks (MOFs) based on their inherent Mie scattering features, including adsorption, thermal decomposition, and expansion[24–26]. Nevertheless, how the structures of local chemical bonds influence the whole flexibility of COFs remains unknown.

In the present work, we describe an unexpected flexibility decrease when imine-based COF-300 with abundant rigid C=N double bonds is reduced to the corresponding secondary amine (C−N bonds) frameworks (COF-300-AR)[27], directly observed by DFM with single-particle resolution (Fig. 1). The generated COF-300-AR well preserves the topological structure and crystallinity of parent COF-300. The significant deformation differences between COF-300 and COF-300-AR are precisely visualized in situ and in real time under exposure to multiple solvent vapours (Fig. 1). Single-particle DFM images disclose that COF-300 crystals are sharply deformed upon interacting with solvent guests, whereas COF-300-AR particles exhibit a very weak host-guest interaction with solvents under a similar experimental condition. DFM imaging in combination with variable-temperature XRD, infrared (IR) spectroscopy, and theoretical simulation allow us to explain the unusual flexible behaviours of COF-300 and COF-300-AR crystals, resulting in the finding of the unique pedal motion of C=N bonds-dependent flexibility at the single-particle level. These results reveal that the introduction of freely-rotating single bonds may worsen rather than improve the flexibility, which provides additional insights into the underlying origin for the flexibility of COFs.

## Results
### Discovery of the flexibility difference between COF-300 and COF-300-AR

In the experiments, we choose COF-300 and COF-300-AR to study the effect of double/single bonds on the flexibility of COFs as COF-300-AR possesses the almost same underlying topology and crystallinity as those of COF-300. According to the reported protocol[27], the COF-300-AR is prepared by chemical reduction of COF-300 (Fig. 2a). Supplementary Figs. 1 and 2 show the scanning electron microscope (SEM) image of COF-300 crystals and the corresponding size distribution. It is clear that the resulting COF-300 particles are highly heterogeneous in both shape and size. The reduction from COF-300 into COF-300-AR does not obviously alter the shape and size (Supplementary Figs. 3 and

4). The successful chemical transformation of imines (C=N double bond) to the second amines (C−N single bond) is confirmed by Fourier transform infrared (FT-IR) spectroscopy. In the FT-IR spectra, the stretching vibration of imine groups at $1614\,cm^{-1}$ disappears, and another two new peaks at $1606\,cm^{-1}$ and $1251\,cm^{-1}$ that are ascribed to N−H bending and C−N stretching vibrations in the secondary amine (C-NH-) groups emerge (Supplementary Fig. 5), demonstrating the conversion of this reduction from COF-300 to COF-300-AR. Meanwhile, X-ray diffraction patterns of COF-300 and COF-300-AR reveal that the crystallinity and the underlying topology are not changed before and after reduction, including space groups (tetragonal I41/a) and lattice parameters (Supplementary Figs. 6 and 7). Meanwhile, both COF-300 and COF-300-AR show Type I isotherm, along with a similar pore volume ($P_v = 0.38\,cm^3/g$ for COF-300 and $P_v = 0.35\,cm^3/g$ for COF-300-AR) (Supplementary Fig. 8). These observations are well consistent with the previously reported data[27], suggesting that the COF-300-AR and COF-300 are isostructural and differ only in the chemical linkages.

Using DFM imaging strategy, we record the movies showing the flexible deformation of single COF-300 or COF-300-AR crystal upon exposure to chloroform vapour as a typical guest molecule (Supplementary Movie 1 and Supplementary Movie 2). Figure 2b, c depicts several snapshots of the two movies at different reaction time ($t$). At $t = 0$ s, both COF-300 and COF-300-AR are ellipsoidal (Fig. 2b, c). As the reaction time prolongs, the individual COF-300 crystal begins to deform measurably (for instance, t = 71 s), while there is no detectable deformation for a single COF-300-AR under the same experimental condition. Finally, a single COF-300 particle sharply deforms and becomes smooth due to solvent-induced crystal expansion (Fig. 2b). Note that the expansion deformation of COF-300 is anisotropic due to its one-dimensional (1D) straight channels along the crystallographic c-axis (Supplementary Fig. 9). To clearly and quantitatively compare the deformation difference, we introduce the roundness (R) to monitor the flexible deformation process (See the definition of R in Supplementary information). The corresponding roundness change (ΔR) is presented in Fig. 2d. As expected, the ΔR of a single COF-300 crystal continuously increases and eventually levels off as the reaction proceeds. In contrast, the ΔR for COF-300-AR is invariable approximately during the reaction. Furthermore, to achieve the definite flexible difference between COF-300 and COF-300-AR, we study multiple COF-300 and COF-300-AR crystals. The particle-averaged deformation behaviors also indicate that COF-300 is much more flexible than COF-300-AR (Fig. 2e and Supplementary Figs. 10 and 11).

### Concentration effect on the flexibility of COF-300 and COF-300-AR

To further identify the flexibility difference between COF-300 and COF-300-AR, the effect of chloroform vapour concentration on the deformation process is also examined. At a low pressure of chloroform (0.79 bar, for example), both COF-300 and COF-300-AR are not

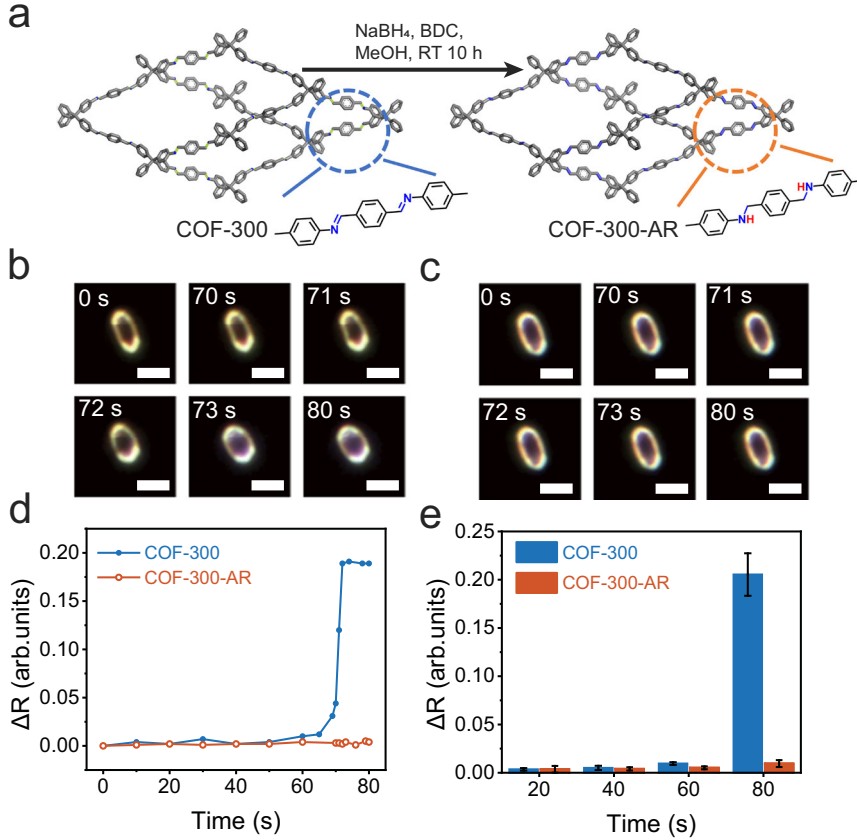

**Fig. 2 | Single-particle DFM imaging of the flexible deformation of COF-300 and COF-300-AR. a** Schematic of the reduction of COF-300 to yield COF-300-AR. Time-lapsed DFM images of (**b**) single COF-300 and **c** COF-300-AR individual in the presence of chloroform vapour (0.84 bar, 25 °C). Scale bars are 4 μm. **d** The corresponding roundness change (ΔR) of single COF-300 and COF-300-AR as a function of reaction time. **e** Particle-averaged ΔR of COF-300 and COF-300-AR crystals under different reaction time. Error bars stand for the standard deviations of ΔR.

deformed, and the ΔR values are zero all the time, as shown in Fig. 3a–c and Supplementary Figs. 12 and 13. However, when the pressure of chloroform vapour is increased to 1.01 bar, COF-300 and COF-300-AR crystals are shortened and rounded in their tip regions. The deformation degree of COF-300 at the equilibrium state is significantly higher than that of COF-300-AR (Fig. 3e and Supplementary Figs. 14 and 15). Note that the average shape alteration of COF-300 is still obviously superior to that of COF-300-AR even if the chloroform liquid is added onto the surface of two COF crystals (Supplementary Figs. 16–18). These experimental results clearly indicate that although the flexible properties of COF-300 demand a concentration-driven process, an obvious decrease in the flexibility is undoubtedly validated when imine-linked COF-300 is reduced to COF-300-AR connected by C–N single bonds.

## Guest property-dependent deformation of COF-300 and COF-300-AR

In addition, to prove the generality of the flexibility difference between COF-300 and COF-300-AR for different guest molecules, we image the corresponding structural evolution processes in the presence of various solvent vapours, including isopropanol (IPA), n-hexane, water, tetrahydrofuran (THF), and ethyl acetate (EA). As displayed in Fig. 4 and Supplementary Figs. 19–30, when introducing other solvent vapours except for water, COF-300 crystals significantly deform, while the structural deformation of COF-300-AR crystals can be ignored. All these results imply that the unique C=N bonds in COFs play a vital role in the regulation of flexibility. It is worthy that the flexibility of COFs depends on not only the structure of the host but also the property of

the guest. Taking water for example, there are no visible shape transformations for COF-300 and COF-300-AR after pumping a water vapour (Fig. 4 and Supplementary Figs. 23 and 24). The reason is attributed to that the hydrophobic internal environment of two COFs blocks the entry of water molecules.

## Inter-particle variations of flexible behaviors of COF-300 crystals

The high spatial resolution and wide-field imaging feature of DFM enable in situ examining single-particle-level deformation processes on highly heterogeneous COF-300 crystals in terms of size and shape, which are inaccessible to conventional bulk-ensemble tools. Figure 5a displays the time-sequential DFM images of six representative COF-300 crystals (marked with P1 to P6) at a 0.84 bar chloroform vapour. Inter-particle heterogeneity in deformations is clear because each COF-300 shows a distinct tendency of ΔR (Fig. 5b). To better access the particle-to-particle heterogeneity, the dΔR curves of each COF-300 crystal are obtained by the first derivative of ΔR with respect to time (Fig. 5c), and the corresponding full width at half maximum (FWHM) is extracted. dΔR and FWHM represent the deformation rate and duration time. For small and irregular COF-300 crystals such as P1, P3, and P4, they firstly deform (Fig. 5b), but the deformation rate is slow, corresponding to a long duration time. On the contrary, the ellipsoidal COF-300 crystals with a large size (P2, P5, and P6) require a longer waiting period prior to the appearance of structural deformation. Once there are enough chloroform guests present in the framework, the deformation process is rapidly completed because of strong internal pressure, showing a high dΔR but low FWHM (Fig. 5d). These

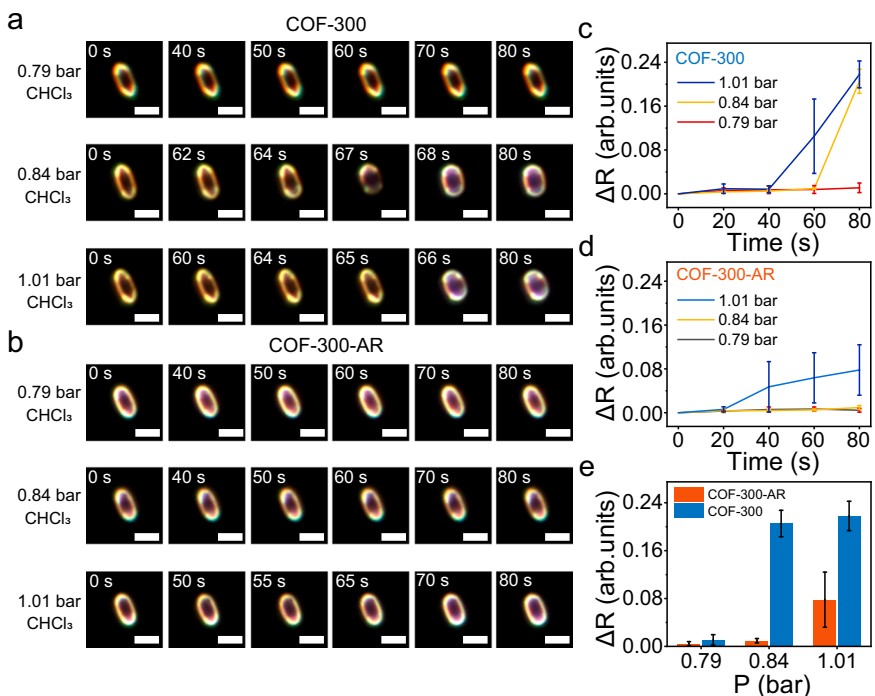

**Fig. 3 | Effect of guest concentration on deformations of two COF crystals.**
**a**–**d** Reaction time series of in situ DFM images and statistical analyses of ΔR for COF-300 and COF-300-AR crystals at three chloroform pressures. Scale bars, 4 μm.

**e** Further comparison of equilibrium ΔR between COF-300 and COF-300-AR under different pressures of chloroform vapours. Error bars stand for the standard deviations of ΔR.

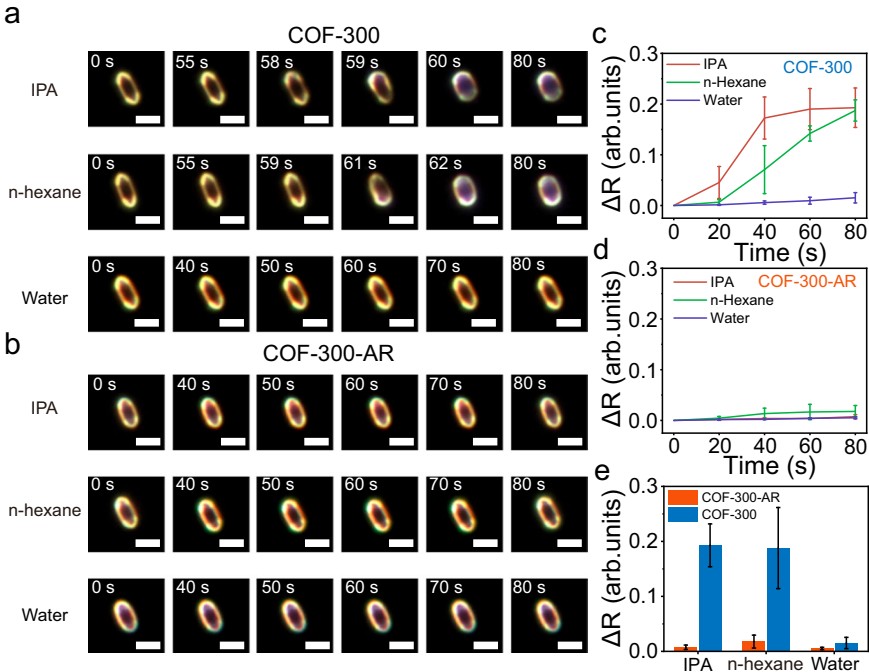

**Fig. 4 | Imaging the flexibility difference in the presence of other guests.**
**a**–**d** Time evolution of DFM images and averaged ΔR for COF-300 and COF-300-AR crystals in the presence of 0.84 bar IPA, 0.81 bar hexane, and 0.69 bar water

vapours. Scale bars, 4 μm. **e** The equilibrium ΔR of COF-300 and COF-300-AR for different solvent guests. Error bars stand for the standard deviations of ΔR.

results suggest a distinct energy barrier to be overcome for deformation among the COF-300 individuals, which is evidenced by theoretical investigations (Supplementary Fig. 31). In fact, the regular and large-sized COFs with a compact composition require a higher internal threshold concentration of guest molecules than that of small and irregular crystals, possessing a long induction time.

## Spectroscopic evidence and X-ray diffraction of pedal motion in COF-300 crystals

Some organic crystals such as azobenzenes, benzylideneanilines, and stilbenes have been shown to undergo a pedal motion, in which bilateral phenyl rings as bicycle pedals rotate around the central double bond (N=N, C=N, or C=C) as the crank arm (Fig. 6e)[28,29]. Previous reports have

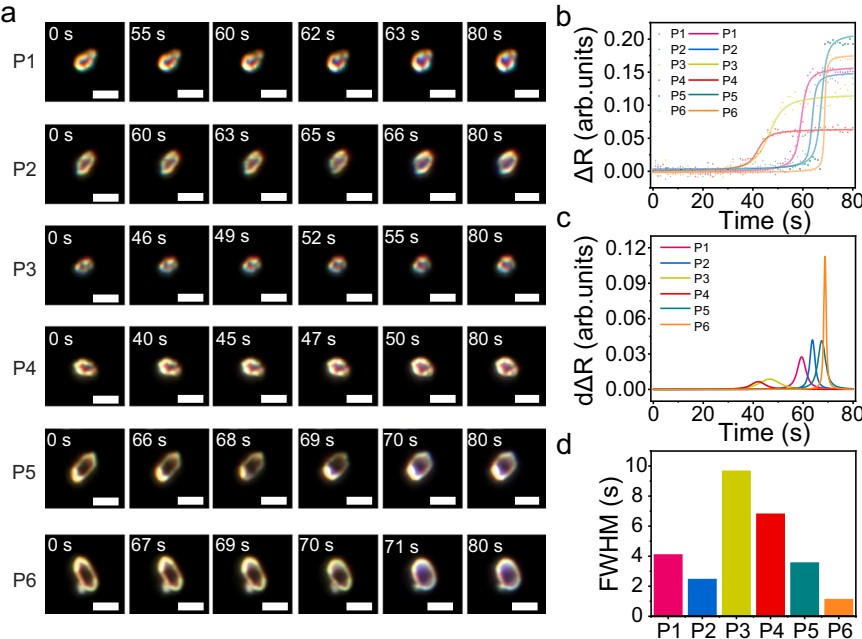

**Fig. 5 | Particle-to-particle variations of deformations on COF-300 crystals.**
**a** Time-sequential DFM images of six representative COF-300 crystals labeled with P1-P6 when exposed to 0.84 bar chloroform at 25 °C. Scale bar = 4 μm. **b** The corresponding ΔR, **c** derivative of ΔR, and **d** full width at half maximum (FWHM) of the curves in (**c**).

demonstrated that the pedal motion can regulate the expansion behavior of organic co-crystals[30]. In consequence, we infer that the flexible deformation of COF-300 crystals is dependent on the pedal motion around rigid C = N bonds in the framework. To determine the dynamic conformations of C=N bonds in COF-300, variable-temperature infrared spectra are measured under a nitrogen atmosphere at different temperatures. As depicted in Fig. 6a and Supplementary Fig. 32, the stretch vibration of C=N bonds at 1613.6 cm$^{-1}$ is shifted to 1618.9 cm$^{-1}$ when increasing the temperature from 300 K to 500 K, illustrating the different conformations of C=N bonds (planar/nonplanar) due to the pedal motion[29]. Oppositely, the N−H bending vibration at 1252 cm$^{-1}$ in the COF-300-AR is almost invariable at different temperatures (Supplementary Fig. 33). The results prove that the molecular motions in COF-300-AR with C−N single bonds are very weak compared with COF-300 because of the lack of pedal motion.

To gain more evidence on the pedal motion within COF-300, we investigate the crystal structures and molecular configurations by variable-temperature XRD. As displayed in Fig. 6b and Supplementary Fig. 34, from 300 to 500 K, there are successive decreases in the diffraction peak positions, suggesting that the COF-300 crystal is expanded. Nevertheless, for COF-300-AR, increasing the temperature results in a very tiny change of the diffraction peak positions (Supplementary Fig. 35), revealing the very weak crystal deformation capacity of COF-300-AR. Further, the imine-linked configurations within COF-300 crystal are simulated by virtue of variable-temperature XRD patterns (Fig. 6c, d and Supplementary Figs. 36–38). Different conformers of the imine linkages in COF-300 are verified at different temperature (Fig. 6c, d), because various conformers from the pedal motion-induced concomitant conformational interconversions have distinct temperature-dependent relative stabilities (Fig. 6e)[28,31]. Other parameters such as bond lengths and bond angles are not changed at different temperature (Supplementary Tables 1 and 2).

### Theoretical analysis of pedal motion
In order to acquire further insight into the pedal motion within COF-300, we conduct quantum-mechanical calculations on N, 1-diphenylmethanimine as a model molecule (Fig. 7a). The torsional angle

potential energy is computed by using a torsional driver to the dihedral angles C(2)-C(1)-N(7)-C(8) of the molecule[29,32]. The existence of the center of symmetry generates an opposite rotation of the C(13)-C(9)-C(8)-N(7) at the torsion angle of 180° in Fig. 7b, c. Two typical conformers (1 and 2) can be converted by torsion motion around the C-C bond (Fig. 7c, d), which undergoes a transition state (3) (Fig. 7d). The overall path results in the pedal motion with respect to the C=N bond. In addition, we further calculate the infrared absorption band of C=N bonds for two conformers and the transition state (Fig. 7d), displaying distinct absorption frequencies (Fig. 7e). These calculated data are well in line with the corresponding results obtained from variable-temperature IR spectra (Fig. 6a). As a result, variable-temperature XRD, IR data and theoretical calculations give a clear indication of the pedal motion in COF-300 crystal.

## Discussion
In summary, we have applied in situ DFM to visually study the flexible deformations of single imine-based COF-300 before and after reduction to COF-300-AR, uncovering an unforeseen and significantly diminished flexibility in COF-300-AR with freely-rotating single bonds. The present imaging results demonstrate that the observed flexible deformations of COF-300 crystals are influenced by guest concentrations and properties, displaying particle-to-particle heterogeneity. By combining DFM observations, variable-temperature XRD/IR measurements, and theoretical calculations, we have revealed that the pedal motion of C=N double bonds dictates the flexible deformations of COF-300. Given the widespread C=N linkages within COFs, the discovered pedal motion-dependent deformation not only enriches our fundamental understanding of the flexibility, but also paves the way for rational design and development of more flexible COFs. From the point of view of accurate measurement, these findings highlight the requirement to employ in situ high-resolution tools for evaluating the flexibility of COFs at the single-particle level.

## Methods
### Materials and reagents
p-phthalaldehyde (BDA), tetrakis(4-aminophenyl)methane (TAM), and aniline were obtained from Leyan Chemical Company. 1, 4-dioxane,

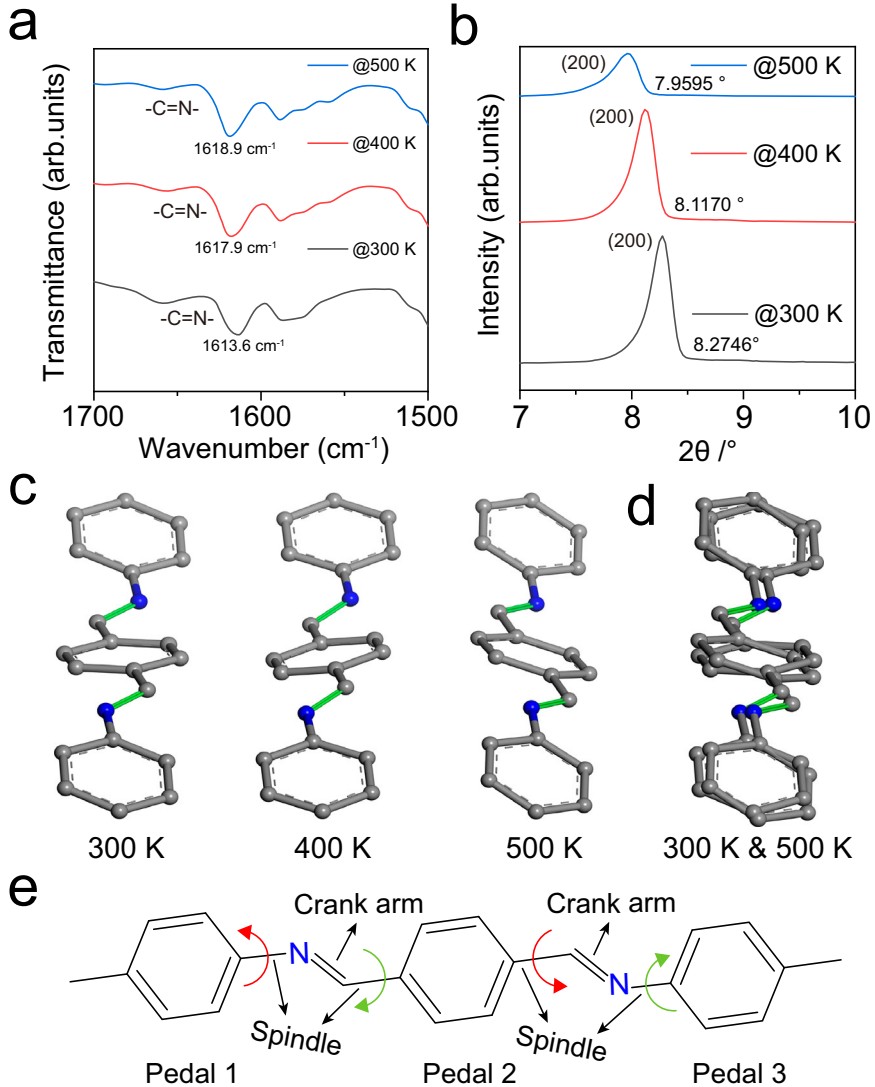

**Fig. 6 | Experimental evidence of pedal motion. a** Magnified variable-temperature IR spectra and **b** XRD patterns of COF-300 under nitrogen atmosphere. **c** The simulated imine-linked configurations in COF-300 under different reaction temperature. **d** The overlaid imine-linked configurations at 300 and 500 K. **e** Pedal motion of imine-linked configuration in COF-300.

acetonitrile, and p-phthalic acid were purchased from Aladdin Reagent Co. Ltd. Sodium borohydride, trichloromethane, dichloromethane, ethyl acetate, tetrahydrofuran, methanol, ethanol, and isopropanol were purchased from Chengdu Kelong Chemical Co., Ltd. All the chemical reagents are of analytical pure grade and directly used without any purification.

### Preparation of COF-300 and COF-300-AR crystals
COF-300 crystals were synthesized by an imine-exchange protocol[18]. Briefly, the mixture of 12.0 mg BDA, 50 μL aniline, 0.5 mL 1,4-dioxane, and 0.2 mL acetic acid (6 M) was added to a glass vial, followed by the injection of 0.5 mL TAM 1,4-dioxane solution (40.0 mgmL$^{-1}$). The reaction mixture is kept at 25 °C for 3.5 days. The final product is collected by centrifugation, washed with ethanol and dried at 70 °C.

COF-300-AR crystals are further prepared by the reduction of COF-300 using NaBH$_4$ as the reductant[27]. Briefly, a mixture consisting of COF-300 crystals (50 mg), p-phthalic acid (57.6 mg), and 25 mL of methanol was prepared at room temperature under stirring for 5 min until COF-300 was fully dispersed. Subsequently, 263.2 mg of NaBH$_4$ is slowly added into the above mixture within 10 min, which was further stirred for 10 h. The light-yellow COF-300-AR crystals were obtained by

centrifugation and washing with deionized water (18.2 MΩ cm) and ethanol several times, followed by drying at 70 °C.

### Experimental setup and single-particle DFM imaging
The dark-field microscopy setup was fabricated on the BX53 microscope (Olympus), on which a halogen lamp incident light source (100 W), an oil-immersion dark-field condenser (UDCW, numerical aperture (N.A.) = 1.2–1.4), a ×20 objective (N.A. = 0.5), and a color charge-coupled-device camera (MicroPublisher 6™, Olympus) were joined. Before single-particle imaging, COF-300 or COF-300-AR particles were dispersed into deionized water under ultra-sonication. After that, 5 μL of COF-300 or COF-300-AR dispersion was added onto a clean glass slide and dried at 25 °C. A coverslip and two cannulas as the inlet and outlet of solvents were then placed onto the glass slide and sealed with glue, which served as the micro-reaction chamber. Eventually, variable solvent vapours or liquids were introduced to the micro-reaction chamber for single-particle DFM imaging.

### Characterizations of COF-300 and COF-300-AR crystals
The morphologies of COF-300 and COF-300-AR crystals were characterized by ZEISS Ultra-55 scanning electron microscope. The

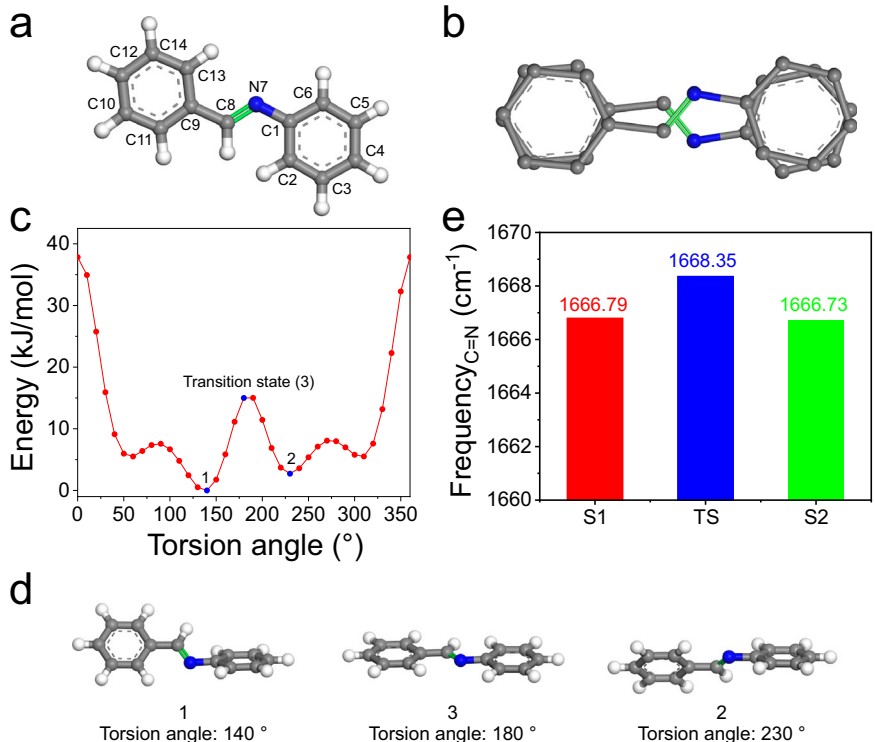

**Fig. 7 | Theoretical investigation of pedal motion. a** Molecular structure and **b** orientational disorder for N, 1-diphenylmethanimine. **c** Torsional angle potential energy profile along the dihedral angles C(2)-C(1)-N(7)-C(8) and **d** the two typical conformers (1 and 2) and transition state (3). **e** The calculated infrared absorption frequencies of 1, 2, and 3.

crystallinity of COF-300 and COF-300-AR was analyzed by X-ray diffraction performed on an X-ray diffractometer (Ultima IV, Rigaku). Fourier transform infrared spectroscopy measurements were carried out on a SPECTRUM ONE AUTOIMA spectrometer. Variable-temperature X-ray diffraction (PANalytical, X'Pert PRO MPD with an Anton Paar accessory) in the temperature range of 300–500 K was used to characterize the crystal configurations of COF-300 and COF-300-AR under a nitrogen atmosphere. Variable-temperature Fourier-transform infrared spectroscopy measurements were measured using a Nicolet iS50 spectrometer equipped with an automated temperature controller (Harrick Scientific) under a nitrogen environment. $N_2$ adsorption-desorption isotherm measurements were performed at 77 K on a Micromeritics ASAP 2460 porosimeter.

## Data availability

The data that support the findings detailed in this study are available within the article and supplementary information as well as on figshare (https://doi.org/10.6084/m9.figshare.23622957). Additional data are available from the corresponding author upon request.

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

## Acknowledgements
We gratefully acknowledge the financial support from the Key Research and Development Project of the Science & Technology Department of Sichuan Province (Grant No. 2021YFSO316). The authors also thank Dr. Mei Tang and Dr. Zhenwei Niu from Southwest University of Science and Technology for theoretical simulations.

## Author contributions
Y.H. conceived and designed the research as well as wrote the manuscript. H.C., Y.L., Z.L., and W.C. conducted the experiments. All authors analyzed the results and commented on the manuscript.

## Competing interests
The authors declare no competing interests.
