## [Peer Review File · Nature Communications]

Direct visual observation of pedal motion-dependent flexibility of single covalent organic frameworksREVIEWER COMMENTS

Reviewer #1 (Remarks to the Author):

In this work, the author compared the flexibility of single COF-300 and its reduced COF-300-AR through the dark-field microscopy. The flexible deformation of the framework decreased after the reduction of the imine bond to the single-bond secondary amine. At the same time, the relationship between the flexible deformation of COF and the concentration and properties of the guest is also directly and visually demonstrated. It will be a significant contribution to this field and will be of interest to a broad readership of Nature Communication. I think it can be accepted after addressing following issues.

1. On page 5, the name of the COF selected in the experiment should be given accurately, which should be "COF-300 and COF-300-AR".
2. When studying the influence of the guest property on the flexible deformation of COF, the hydrophobic of the framework causes no obvious deformation of the two materials in the water vapor. However, will the two different bond types, such as C=N double bond and C-N single bond in the framework, lead to different affinity of COF-300 and COF-300-AR to other solvent vapors?
3. There are some errors in the article that the picture label is inconsistent with the article content. For example, the variable-temperature XRD shown in Figure 5 and Figure S32 are from 300 K to 500 K, while the corresponding description in the article ranges from 100 K to 300 K. It is better to keep the pictures and article consistent.
4. There are some spelling mistakes in the article, please check it carefully. For example, the caption in Figure S35, "mine-linked" should be "imine-linked".

Reviewer #2 (Remarks to the Author):

While flexible, "breathing" metalorganic frameworks have been studied intensively, this phenomenon has been observed to a much lesser extent with covalent organic frameworks (COFs). Here, the authors aim to shed light on the direct correlation between the structure of a COF lattice and its flexible behavior. To this end, they investigate the flexible behavior of the well-known 3D COFs COF-300 and COF-300-AR "live" with dark-field microscopy (enabling the observation of individual crystals in the heterogeneous ensemble) as a

function of being exposed to different solvent vapors (concentration-dependent) such as chloroform. The two topologically similar COFs differ at their linkage between the tetrahedral tetraphenyl methane and the linear phenyl moieties. In COF-300, the moieties are linked by rather rigid imine linkages, while the latter were reduced (with NaBH₄) to nearly freely rotating secondary amine linkages in COF-300-AR.

The authors report that unexpectedly, the solvent-induced deformation of the imine-linked COF turned out to be much more substantial (and could be achieved with different solvents) than that of the amine-linked COF containing a high concentration of single bonds. Based on variable-temperature X-ray diffraction experiments and aided by variable-temperature infrared spectroscopy data the authors advance a model for this unexpected behavior, proposing that the deformation of COF-300 is due to a pedal motion around the rigid imine double bonds. The authors suggest that a distinct energy barrier must be overcome for the pedal motion to proceed.

Technical points and issues

- Crystallinity of the reduced form is worse and diffraction patterns differ quite substantially, please explain. How can the proposed unit cell dimensions be so similar in both cases when the C-N bond length differs substantially from that of C=N?
- How can the effect of heating on the lattice be related to that of solvent adsorption?
- Please show with XRD data if the deformation is really reversible.
- Discuss the impact of interpenetration on flexibility.
- Add nitrogen adsorption isotherms of the two COFs.
- In the absence of detailed experimental evidence, the study requires a theoretical investigation of the proposed energy barrier and pedal motion.
- How can the minor shifts of the 200 reflection (Fig. 5) cause the large shape changes observed in the optical micrographs? How is the anisotropic deformation of the crystals explained? Need to propose a structural model.
- Style needs to be improved.

Summarizing, while this is an original study of potential interest to the COF community, the above technical points need to be addressed before publication can be recommended. A

major revision is required.

Reviewer #3 (Remarks to the Author):

In this work, the authors synthesized COF-300-AR with freely-rotating C-N bonds by the reduction of imine-based COF-300 consisting of rigid C=N bonds, and used DFM to investigate the flexible deformation of COF-300 and COF-300-AR. Overall, this work is a little more interesting, but the content is quite similar with the authors' previous work (Angew. Chem. Int. Ed. 2023, 62, e202214569), leading to the lack of novelty. Combining with the following issues, I don't think this manuscript meets the standard for publication in Nature Communications.

1. The deformations of COF-300 have been comprehensively studied in the previous work, but the time for COF-300 crystal beginning to deform in this work (71 seconds) is obviously longer than that in the previous report (only a few seconds). What causes such a difference?
2. Although the authors investigated how the structures of local chemical bonds influence the whole flexibility, the mechanism is only based on some previous reports (Chem. Soc. Rev. 2009, 38, 2244-2252; Angew. Chem. Int. Ed. 2018, 57, 1792-1796; J. Am. Chem. Soc. 2001, 123, 10884-10888) at the initial stage without an in-depth study. Theoretical calculation is recommended to confirm the mechanism.
3. The XRD patterns of COF-300 and COF-300-AR are significantly different, so the claim "the crystallinity and the underlying topology are not changed before and after reduction" is not accurate. In addition, the diffraction peak position and intensity in XRD patterns are not consistent with the previously reported data by other researchers in Ref. 27. Does the crystallinity cause some influence on the discovery of flexibility?
4. Some mistakes should be corrected, such as "shed" in line 39, "from 100 K to 300 K" in line 250.

Response to the referee comments:

Reviewer 1:

Q1. On page 5, the name of the COF selected in the experiment should be given accurately, which should be "COF-300 and COF-300-AR".

Thank you for your constructive and important suggestion. According to the suggestion, the name of the COF in the experiment has been revised to "COF-300 and COF-300-AR" in our revised manuscript marked in red.

Q2. When studying the influence of the guest property on the flexible deformation of COF, the hydrophobic of the framework causes no obvious deformation of the two materials in the water vapor. However, will the two different bond types, such as C=N double bond and C-N single bond in the framework, lead to different affinity of COF-300 and COF-300-AR to other solvent vapors?

The affinity between COF-300 and solvents is usually ascribed to the noncovalent interactions such as van der Waals forces and hydrogen bonding (*Science* 2018, 361, 48-52). It is expected that COF-300-AR with C-N-H groups have a higher affinity than that of COF-300 because of the increase in molecule weight and electron density of N atoms after reduction. However, the observed flexible deformation of COF-300 is extremely higher than that of COF-300-AR, essentially ruling out the different affinity-caused flexible deformation of two COFs.

Q3. There are some errors in the article that the picture label is inconsistent with the article content. For example, the variable-temperature XRD shown in Figure 5 and Figure S32 are from 300 K to 500 K, while the corresponding description in the article ranges from 100 K to 300 K. It is better to keep the pictures and article consistent.

According to the suggestion, we revise the related description in our revised manuscript marked in red.

Q4. There are some spelling mistakes in the article, please check it carefully. For example, the caption in Figure S35, "mine-linked" should be "imine-linked".

According to the suggestion, we check the spelling of this manuscript word by word. Some mistakes have been corrected in our revised manuscript.

Reviewer 2:

Q1. Crystallinity of the reduced form is worse and diffraction patterns differ quite substantially, please explain. How can the proposed unit cell dimensions be so similar in both cases when the C-N bond length differs substantially from that of C=N?

Thank you for your valuable and positive suggestions. According to the suggestion, we check the COF-300-AR sample carefully by DFM. As shown in Figure 1, there are some white crystals within the COF-300-AR, which are attributed to the residual p-phthalic acid (Experiment section in our manuscript). The impurity contributes many diffraction patterns, leading to the difference in the diffraction patterns between COF-300 and COF-300-AR.

Figure 1. DFM images of COF-300-AR samples.

To obtain the real diffraction patterns of COF-300-AR, the COF-300-AR samples are purified by using 0.1 M NaOH washing treatments. As illustrated in Figure 2, PXRD patterns of COF-300-AR are in good agreement with that of COF-300. Moreover, based on the Pawley refinement, the same tetragonal $I4_1/a$ space group is discovered in COF-300 and COF-300-AR crystals (Figure 3 and Figure 4), along with a slight lattice parameter change due to the change in the hybridization of N and C atoms from sp^2 to sp^3 (*Chem*, 2018, 4, 1696-1709). These results confirm that the crystallinity and the underlying topology are well preserved in the chemical transformation from COF-300 to COF-300-AR.

Figure 2. PXRD patterns of COF-300 and purified COF-300-AR.

It should be noted that the existence of p-phthalic acid does not affect the obtained single-particle imaging results because DFM with a high spatial resolution can easily distinguish COF-300-AR from p-phthalic acid. However, as a common bulk-ensemble tool, XRD characterizations do not have the capacity.

Figure 3. XRD pattern of COF-300 crystals (red), Pawley refined (black), and

difference plot (blue).

Figure 4. XRD pattern of COF-300-AR crystals (red), Pawley refined (black), and difference plot (blue).

According to the suggestion, the updated XRD data has been added to our revised manuscript (Supplementary Figure 6 and Figure 7 in our revised manuscript).

Q2. How can the effect of heating on the lattice be related to that of solvent adsorption?

The solvent adsorption of COF-300 induces the change of the lattice, which has been previously demonstrated (J. Am. Chem. Soc. 2019, 141, 3298-3303). In this work, we attribute the lattice change to the pedal motion-induced solvent adsorption. The covalent bond motion is related to the temperature. In order to confirm the pedal motion, we measure the XRD patterns and FTIR spectra under different reaction temperature.

Q3. Please show with XRD data if the deformation is really reversible.

After carefully comparing the corresponding DFM images shown in Figure 5, there is a difference in the morphology between empty COF-300 and chloroform-filled COF-300 after evacuation, especially in tip regions, suggesting that the deformation is not really reversible. According to the suggestion, we delete the related descriptions in our revised manuscript.

Figure 5. DFM images of single empty COF-300 and chloroform-filled COF-300 before and after evacuation.

Q4. Discuss the impact of interpenetration on flexibility.

According to the suggestion, we theoretically analyze the impact of interpenetration on flexibility because COF-300 with a 5-fold interpenetrated diamond topology (*dia-c5*) is not stable (*J. Am. Chem. Soc.* 2018, 140, 6763-6766). The bulk modulus and Yong's modulus for the 7-fold interpenetrated diamond structure of COF-300 (*dia-c7*) and *dia-c5* COF-300 are calculated by density functional theory as shown in Figure 6. It is evident that the values of K_v and E_v for *dia-c5* COF-300 are slightly higher than those of its corresponding interpenetration isomerism (*dia-c7* COF-300), suggesting the relatively weak flexibility of *dia-c5* COF-300.

Figure 6. (a) Schematic illustration of 5-fold and 7-fold interpenetrated diamond topology structures for two COF-300 frameworks (*dia-c5* and *dia-c7*). (b, c) The calculated bulk modulus (K_v) and Young's modulus (E_v) for *dia-c7* COF-300 and *dia-c5* COF-300.

Q5. Add nitrogen adsorption isotherms of the two COFs.

According to the suggestion, the nitrogen adsorption isotherms of COF-300 and COF-300-AR (Figure 7) have been added and discussed in our revised manuscript. It can be seen from Figure 7 that both COF-300 and COF-300-AR show Type I isotherm, along with a similar pore volume ($P_v = 0.38 \text{ cm}^3/\text{g}$ for COF-300 and $P_v = 0.35 \text{ cm}^3/\text{g}$ for COF-300-AR). The corresponding Brunauer-Emmett-Teller (BET) surface areas are calculated to be $1261.6 \text{ m}^2/\text{g}$ and $886.4 \text{ m}^2/\text{g}$ for COF-300 and COF-300-AR, respectively. The decrease in the surface area of COF-300-AR can be attributed to an increase in framework mass or the generation of amorphous oligomers presented in pores after reduction (*J. Am. Chem. Soc.* 2022, 144, 1138-1143). The spatial resolution

capacity of the present single-particle imaging method can obtain the COF-300 and COF-300 with similar shape and crystallinity to accurately investigate the pedal motion-dependent flexibility.

Figure 7. N_2 adsorption-desorption measurements of COF-300 and COF-300-AR.

The related Figures, descriptions, and references have been added to our revised supporting information (Supplementary Figure 8 in our revised manuscript).

Q6. In the absence of detailed experimental evidence, the study requires a theoretical investigation of the proposed energy barrier and pedal motion.

According to the suggestion, we theoretically study the proposed energy barrier and pedal motion within COF-300 framework. Based on the first-principles density functional theory and Monte Carlo simulations, a ‘gate-opening’ model for interpreting the guest-induced pore expansion is fabricated, which is adopted the model proposed by Kitagawa et al (*J. Am. Chem. Soc.* 2017, 139, 18313-18321; *Nat. Material.* 2023, 22, 636-643). The deformation energy (E_{def}) and interaction energy (E_{int}) stand for the

energy difference between closed- and open-pore states as well as the energy decrease after the sorption of CH_3Cl . The sum of E_{def} and E_{int} is the total energy (E_{tot}). As displayed in Figure 8, the E_{def} is about $17.9 \text{ kcal mol}^{-1}$ for the sorption of four CHCl_3 , and it varies to $70.1 \text{ kcal mol}^{-1}$ after adsorbing sixteen CHCl_3 , clarifying the requirement of the energy barrier for triggering the pedal motion. The related descriptions have been added to our revised supporting information (Supplementary Figure 31).

Figure 8. Calculation of E_{def} , E_{int} , and E_{tot} during the sorption of different numbers of CHCl_3 .

In order to acquire some insight into the pedal motion of COF-300, we conduct quantum-mechanical calculations on N, 1-diphenylmethanimine as a model molecule (Figure 9). The energy profile is computed by using a torsional driver to the dihedral angles C(2)-C(1)-N(7)-C(8) of the molecule (J. Am. Chem. Soc. 1999, 121, 3767-3772 3767; Angew. Chem. Int. Ed. 2018, 57, 1792-1796). The existence of the center of symmetry generates an opposite rotation of the C(13)-C(9)-C(8)-N(7) at the torsion angle of 180° in Figure 9b and Figure 9c. Two typical conformers (1 and 2) can be converted by torsion motion around the C-C bond (Figure 9c and 9d), which undergoes a transition state (3) (Figure 9d). The overall path results in the pedal motion with respect to the C=N bond. In addition, we further calculate the infrared absorption band

of C=N bonds for two conformers and the transition state (Figure 9d), displaying distinct absorption frequencies (Figure 9e). These calculated data are well in line with the corresponding results obtained from variable-temperature IR spectra (Figure 5a in our revised manuscript).

Figure 9. (a) Molecular structure and (b) orientational disorder for N, 1-diphenylmethanimine. (c) Torsional angle potential energy profile along the dihedral angles C(2)-C(1)-N(7)-C(8) and (d) the two typical conformers (1 and 2) and transition state (3). (e) The calculated infrared absorption frequencies of 1, 2, and 3.

The related descriptions have been added to our revised manuscript marked in red (Fig. 6).

Q7. How can the minor shifts of the 200 reflection (Fig. 5) cause the large shape changes observed in the optical micrographs? How is the anisotropic deformation of the crystals explained? Need to propose a structural model.

According to the suggestion, a structural model is proposed in our revised manuscript. As illustrated in Figure 10, COF-300 has a 7-fold interpenetrated diamond topology in the tetragonal system and one-dimensional (1D) straight channels along the

crystallographic c-axis (Nat. Commun. 2020, 11, 6128). Upon the adsorption of guest molecules, the adamantane-like cages of COF-300 uniformly change themselves along the a and b axes to yield an expanded structure (Figure 10). Although the COF-300 is obviously deformed after the adsorption of guest molecules, the interlayer space along the c-axis direction is slightly enlarged, resulting in the minor shifts of the (200) reflection (Figure 10). Moreover, the 1D channels along the crystallographic c-axis can accommodate large quantities of guest molecules, whereas other directions are not accessible to the guest molecules. Therefore, the inherent channel characteristics in COF-300 lead to the anisotropic deformation of the crystals.

Figure 10. Schematic diagram of the contraction and expansion of COF-300 with 7-fold interweaving upon adsorption and desorption of guest molecules.

The related descriptions have been added to our revised manuscript (Supplementary Figure 9).

Q8. Style needs to be improved.

According to the suggestion, the styles have been improved as far as possible in our revised manuscript.

Reviewer 3:

Q1. The deformations of COF-300 have been comprehensively studied in the

previous work, but the time for COF-300 crystal beginning to deform in this work (71 seconds) is obviously longer than that in the previous report (only a few seconds). What causes such a difference?

Thank you for your comments. Two reasons can cause such a difference. First, the COF size of this work is smaller than that of the previous work, leading to a decrease in interaction sites between COF-300 and chloroform. Second, the difference in shape, crystallinity, and surface defect will also alter the initial deformation, which is demonstrated by DFM imaging (Figure 4 in our manuscript). As a result, the size, shape, crystallinity, and surface defect can not easily be controlled precisely in each experiment, resulting in particle-to-particle variations and emphasizing the significance of single imaging experiments.

Q2. Although the authors investigated how the structures of local chemical bonds influence the whole flexibility, the mechanism is only based on some previous reports (Chem. Soc. Rev. 2009, 38, 2244-2252; Angew. Chem. Int. Ed. 2018, 57, 1792-1796; J. Am. Chem. Soc. 2001, 123, 10884-10888) at the initial stage without an in-depth study. Theoretical calculation is recommended to confirm the mechanism.

Similar to the Q6 of Reviewer 2, according to the suggestion, theoretical calculations are performed to confirm the mechanism (Figure 9). The corresponding descriptions have been added to our revised manuscript (Figure 6 in the manuscript).

Q3. The XRD patterns of COF-300 and COF-300-AR are significantly different, so the claim “the crystallinity and the underlying topology are not changed before and after reduction” is not accurate. In addition, the diffraction peak position and intensity in XRD patterns are not consistent with the previously reported data by other researchers in Ref. 27. Does the crystallinity cause some influence on the discovery of flexibility?

Similar to the Q1 of Reviewer 2, it is found that the residual p-phthalic acid in COF-300-AR samples results in a significant difference in XRD patterns (Figure 1). When the COF-300-AR samples are purified by using 0.1 M NaOH washing treatments, there is a good agreement in XRD patterns between COF-300 and COF-300-AR (Figure 2).

Meanwhile, the Pawley refinement results demonstrate that the crystallinity and the underlying topology are well preserved in the chemical transformation process from COF-300 to COF-300-AR (Figure 3 and Figure 4).

As a result, we can exclude the possibility that the crystallinity causes some influence on the discovery of flexibility.

Q4. Some mistakes should be corrected, such as “shed” in line 39, “from 100 K to 300 K” in line 250.

According to the suggestion, these mistakes have been corrected in our revised manuscript marked in red. The “shed” has been revised to “provide”. The description “from 100 K to 300 K” has been corrected to “from 300 K to 500 K”.

REVIEWERS' COMMENTS

Reviewer #1 (Remarks to the Author):

In the revised manuscript, the authors have provided comprehensive responses to all inquiries raised by the referees. Based on this, I confidently recommend the paper for publication without any further revisions.

Reviewer #2 (Remarks to the Author):

In their revised version of the paper, the authors have addressed almost all technical points and issues raised in the previous review, with a particular focus on detailed theoretical modeling.

One point remains: the nitrogen sorption isotherms of the two COFs are far from equilibrium, they need to be re-measured after better extraction of the samples and using suitable parameters to allow for equilibration. The paper is recommended for publication after minor revision.

Reviewer #3 (Remarks to the Author):

The authors have substantially revised their manuscript based on the comments and suggestions of the reviewers. I have no further comments at this stage.

Response to the referee comments:

Reviewer 1:

Q1. In the revised manuscript, the authors have provided comprehensive responses to all inquiries raised by the referees. Based on this, I confidently recommend the paper for publication without any further revisions.

Thank you for your help.

Reviewer 2:

Q1. In their revised version of the paper, the authors have addressed almost all technical points and issues raised in the previous review, with a particular focus on detailed theoretical modeling. One point remains: the nitrogen sorption isotherms of the two COFs are far from equilibrium, they need to be re-measured after better extraction of the samples and using suitable parameters to allow for equilibration. The paper is recommended for publication after minor revision.

Thank you for your valuable and positive suggestions. In fact, we repeat measurements for the nitrogen sorption isotherms of the two COFs many times. The obtained data is shown in Figure 1 and Supplementary Figure 8 in our revised manuscript. It seems that the sorption isotherms are not equilibrium because the desorption curve does not overlap well with the sorption process. The main reason is that the flexibility of two COFs results in the pore contraction, preventing the desorption of nitrogen.

Figure 1. N₂ adsorption-desorption measurements of COF-300 and COF-300-AR in the present work.

Moreover, to confirm the effectiveness of the present experimental data, we collected the N₂ adsorption-desorption measurements for COF-300 and other flexible COFs in the reported literatures. As shown in Figure 2, many reported nitrogen sorption isotherms for COF-300 are not close, which is quite consistent with our findings. Furthermore, other flexible COFs such as COF-506-Cu and 3D-CageCOF-1 also show a similar nitrogen sorption isotherm to that of COF-300, revealing the flexibility-induced nonequilibrium nitrogen adsorption-desorption process.

Adv. Funct. Mater. 2023, 33, 16, 2300219

Nat. Commun. 2020, 11, 6128

J. Am. Chem. Soc. 2019, 141, 7, 3298-3303

Adv. Mater. Interfaces. 2022, 9, 2201263.

J. Am. Chem. Soc. 2009, 131, 13, 4570-4571

J. Am. Chem. Soc. 2023, 145, 4, 2544-2552

J. Am. Chem. Soc. 2018, 140, 47, 16015-16019

J. Am. Chem. Soc. 2020, 142, 39, 16842-16848

Figure 2. N₂ adsorption-desorption measurements of COF-300 and other flexible COFs in the reported literatures.

According to the suggestion, the corresponding discussions are added to our re-revised supporting information (the bottom of Supplementary Figure 8).

Reviewer 3:

The authors have substantially revised their manuscript based on the comments and suggestions of the reviewers. I have no further comments at this stage.

Thank you for your constructive comments.